# ADAR1 Isoforms Regulate *Let-7d* Processing in Idiopathic Pulmonary Fibrosis

**DOI:** 10.3390/ijms23169028

**Published:** 2022-08-12

**Authors:** Gabriela Díaz-Piña, Karla Rubio, Rosa M. Ordoñez-Razo, Guillermo Barreto, Eduardo Montes, Carina Becerril, Alfonso Salgado, Héctor Cabrera-Fuentes, Arnoldo Aquino-Galvez, Angeles Carlos-Reyes, Victor Ruiz

**Affiliations:** 1Molecular Biology Laboratory, Pulmonary Fibrosis Department, National Institute of Respiratory Diseases “Ismael Cosío Villegas”, Calzada de Tlalpan No. 4502, Col. Sección XVI, Mexico City 14080, Mexico; 2Université de Lorraine, CNRS, Laboratoire IMoPA, UMR 7365, F-54000 Nancy, France; 3Lung Cancer Epigenetics, Max Planck Institute for Heart and Lung Research, 61231 Bad Nauheim, Germany; 4International Laboratory EPIGEN, Consejo de Ciencia y Tecnología del Estado de Puebla (CONCYTEP), Universidad de la Salud del Estado de Puebla, Puebla 72000, Mexico; 5Medical Research Unit in Human Genetics, Pediatric Hospital “Dr. Silvestre Frenk Freud”, National Medical Center “Siglo XXI”, Mexican Social Security Institute, Av. Cuauhtémoc No. 330, Col. Doctores, Delegación Cuauhtémoc, Mexico City 06720, Mexico; 6Cellular Biology Laboratory, Pulmonary Fibrosis Department, National Institute of Respiratory Diseases “Ismael Cosío Villegas”, Calzada de Tlalpan No. 4502, Col. Sección XVI, Mexico City 14080, Mexico; 7Rheumatic Diseases Research Laboratory, Pulmonary Fibrosis Department, National Institute of Respiratory Diseases “Ismael Cosío Villegas”, Calzada de Tlalpan No. 4502, Col. Sección XVI, Mexico City 14080, Mexico; 8Medical School, Institute of Biochemistry, Justus-Liebig-University, 35390 Giessen, Germany; 9Onco-Immunobiology Laboratory, Department of Chronic-Degenerative Diseases, National Institute of Respiratory Diseases “Ismael Cosío Villegas”, Calzada de Tlalpan No. 4502, Col. Sección XVI, Mexico City 14080, Mexico

**Keywords:** ADAR, RNA edition, miRNA, IPF

## Abstract

Double-stranded RNA adenosine deaminase 1 (ADAR1) is significantly down-regulated in fibroblasts derived from Idiopathic Pulmonary Fibrosis (IPF) patients, and its overexpression restored levels of *miRNA-21*, *PELI1*, and *SPRY2*. There are two ADAR1 isoforms in humans, ADAR1-p110 and ADAR1-p150, generated by an alternative promoter. *Let-7d* is considered an essential microRNA in Pulmonary Fibrosis (PF). In silico analysis revealed *COL3A1* and *SMAD2*, proteins involved in the development of IPF, as *Let-7d* targets. We analyzed the role of ADAR1-p110 and ADAR1-p150 isoforms in the regulation of *Let-7d* maturation and the effect of this regulation on the expression of *COL3A1* and *SMAD2* in IPF fibroblast. We demonstrated that differential expression and subcellular distribution of ADAR1 isoforms in fibroblasts contribute to the up-regulation of *pri-miR-Let-7d* and down-regulation of mature *Let-7d*. Induction of overexpression of ADAR1 reestablishes the expression of *pri-miR-Let-7d* and *Let-7d* in lung fibroblasts. The reduction of mature *Let-7d* upregulates the expression of *COL3A1* and *SMAD2*. Thus, ADAR1 isoforms and *Let-7d* could have a synergistic role in IPF, which is a promising explanation in the mechanisms of fibrosis development, and the regulation of both molecules could be used as a therapeutic approach in IPF.

## 1. Introduction

Idiopathic Pulmonary Fibrosis (IPF) is the most frequent and aggressive interstitial lung disease. It is considered a consequence of several risk factors such as smoking habits, genetic determinants, environmental factors, gastroesophageal reflux, viral infections, and aging; however, its origin and further progression are not fully understood [1,2,3,4,5]. In situ, fibroblasts differentiate into myofibroblasts and form fibroblastic foci, which are responsible for secreting excessive amounts of extracellular matrix (ECM) components such as fibrillar collagens [6,7,8]. The excessive accumulation of these proteins entails the destruction of the pulmonary architecture [1,2,3,4,5].

The molecular mechanisms of IPF have not been fully described. Some evidence suggests the possible participation of some microRNAs [9,10,11,12,13]. Several editing microRNAs by Adenosine Deaminase acting on RNA (ADAR) are affected in their maturation process mediated by Drosha and Dicer, or they are redirecting to new mRNAs target because their seed region nucleotides are modified [14,15,16,17,18,19,20,21,22,23]. This edition represents a specific tissue regulatory mechanism of microRNA expression [22].

ADAR1 has two isoforms generated by the alternative promoter, ADAR1-p110 and ADAR1-p150. ADAR1-p110 is the amino-terminally truncated isoform located in nuclear regions and constitutively expressed. ADAR1-p150 is the full-length isoform, which is interferon-inducible and is preferentially located in cytoplasmic regions [14,21]. ADAR1 isoforms have 2–3 double-stranded RNA binding domains (dsRNA) in the amino-terminal region, followed by a deaminase domain in the carboxyl-terminal region and defined domains of subcellular localization that allow them to shift between the nucleolus, nucleus, and cytoplasm [19]. Sub-cellular location differences suggest that ADAR1 isoforms display differential RNA regulation [3]. ADAR1 is significantly down-regulated in IPF fibroblasts, and the over-expression of ADAR1 and ADAR2 reestablishes the expression levels of *miRNA-21*, *PELI1*, and *SPRY2* in fibroblasts from IPF patients [14]. Different works showed a *Let-7d* down-regulation in lung fibroblasts of IPF patients [9,10,12]. *SMAD2* and 3, positive regulator of the TGF-β1 pathway, represents one of the main *Let-7d* targets, while *Let-7d* low expression promotes an increase in mesenchymal markers such as HMGA2, vimentin, N-cadherin, and ACTA2 [10,13]. In leukemia stem cells, ADAR1 can edit *pri-miRNA-Let-7d* transcript forming a secondary double-stranded structure that induces a negative regulation of *Let-7d* and self-renewal [23].

Dysregulation of the microRNA maturation processing caused by ADAR1 editing, and its effects on pulmonary fibroblasts, are not well known. We studied the role of the ADAR1-p110 and ADAR1-p150 isoforms in the regulation of *Let-7d* maturation and the effect of these modifications on the expression of *collagen-3*, a putative *Let-7d* target related to IPF.

## 2. Results

### 2.1. Expression of ADAR1 Isoforms in Controls and IPF Fibroblasts

ADAR1-p110 isoform in IPF fibroblasts (*n* = 10) was downregulated (** *p* = 0.008, Figure 1A), whereas ADAR1-p150 isoform was overexpressed (*** *p* = 0.00054; Figure 1B), compared with control fibroblasts (*n* = 10). Additionally, we verified that there were no differences in the basal levels of ADAR1 expression between both control and fibrotic fibroblasts (Appendix A). We further confirmed these observations at the protein level by Western blot analysis, and we observed that ADAR1-p110 isoform levels were less than 66% (* *p* = 0.05; Figure 1C,D) in total protein extracts in IPF fibroblasts compared to controls. The ADAR1-p150 isoform did not show changes (Figure 1C,E). Different distribution of fluorescence intensity of ADAR1 was detected in the nuclei of the IPF fibroblasts (*n* = 6) compared to control cells (*n* = 6) (Figure 2A). ADAR1 was in the nucleolar region mainly in control fibroblasts, whereas in IPF cells, ADAR1 was exclusively detected in the nucleoplasm Figure 2A,B). These preliminary observations suggested an altered subnuclear location of ADAR1 in IPF. It was confirmed by the analysis of fluorescence intensity. The ratio of ADAR1 nucleolar to nuclear expression is significantly higher in control fibroblasts than in IPF fibroblasts (Figure 2C). 

### 2.2. ADAR1 Is Hyperactive in IPF Fibroblasts

RESS-qPCR analysis [24] was performed to determine RNA editing mediated by ADAR1 in lung fibroblasts. We analyzed six controls and six IPF fibroblasts; IPF fibroblasts possess increased editing activity in specific adenosine of *APOBEC3D* compared with control fibroblasts (* *p* = 0.0193; Figure 3). The editing activity on *APOBEC3D* in control and fibrotic fibroblasts transfected with ADAR1 isoforms showed that the catalytic activity is increased after transfection with active plasmids. In contrast, the mutant plasmids do not modify the activity. In control fibroblasts (black bars), ADAR1-p110 wt shows a significant increase in editing activity (** *p* = 0.003737). A remarkable increase in editing activity is observed with the ADAR1-p150 isoform (** *p* = 0.000139), while the most significant increase the editing activity in IPF fibroblasts are observed after overexpression of the ADAR1-p110 (** *p* = 0.007030). An increase in ADAR1-p150 wt activity was also significant (** *p* = 0.009158). Editing-deficient ADAR-p110 mt and ADAR-p150 mt constructs did not affect the editing activity. ADAR editing activity on healthy control and fibrotic fibroblasts considerably decreases upon Pentostatin treatment, which is a specific inhibitor of the editing activity of Adenosine Deaminases (ADA) (Figure 4). Pentostatin, as a specific ADA inhibitor, decreases the catalytic activity in control fibroblasts (** *p* = 0.0059) and IPF fibroblasts (** *p* = 0.0014). Over-expression of ADAR1-p150 wt (** *p* = 0.0058) and ADAR1-p110 wt (** *p* = 0.0018) increased the catalytic activity of ADAR in control fibroblasts. On the contrary, the overexpression of ADAR1-p150 (*p* = 0.8) and ADAR1-p110 mutant (*p* = 0.72) did not induce significant changes in the catalytic activity in the control fibroblasts. However, ADAR1-p150 wt over-expression in IPF fibroblasts increases its activity (** *p* = 0.0042), similar to the overexpression of ADAR1-p110 (** *p* = 0.0067). Over-expression of ADAR1 mutant plasmids does not modify the activity. ADAR1-p150 mutant (*p* = 0.2) and ADAR1-p110 mutant (*p* = 0.02).

### 2.3. Basal Expression of Let7d and pri-miR-Let7d in Normal and IPF Fibroblasts and Correlation with ADAR1p110 and p150

The basal expression of *pri-miR-Let7d* is higher in IPF fibroblasts than in normal fibroblasts (*p* = 0.05, Figure 5A). On the other hand, *Let7d* expression is lower in IPF fibroblasts than in Normal fibroblasts (*p* = 0.0028, Figure 5D). The correlation of ADAR1 isoforms does not show significant differences between ADAR1-p110 (Figure 5B) and ADAR1-p150 (Figure 5C). Let7d displayed a negative correlation with ADAR1-p150 (*p* < 0.0001) (Figure 5E) and a positive correlation with ADAR1-p110 (*p* = 0.0014) (Figure 5F).

### 2.4. The Effect of ADAR1 Isoforms on Let-7d

To determine the functional role of ADAR1 isoforms in fibroblasts, we transfected control and IPF fibroblasts with ADAR1-p110 wt, ADAR1-p150 wt, and their respective mutated vectors to analyze the levels of primary and mature *Let-7d*. ADAR1-p110 wt transfection in control fibroblasts favored the decrease in *pri-miR-Let-7d* (*p* = 0.054479, Figure 6A), whereas in IPF fibroblasts, both isoforms promoted a significant decrease in *pri-miR-Let-7d*, (ADAR1-p110 wt, ** *p* = 0.0002108, and ADAR1-p150 wt, ** *p* = 0.00038554, Figure 6A). Isoforms over-expression favored the increase in *Let-7d* in control (ADAR1-p110 wt, ** *p* = 0.001477 and ADAR1-p150 wt, ** *p* = 0.001946, Figure 6B) and in fibrotic fibroblasts (ADAR1-p110-wt, *** *p* = 0.000151 and ADAR1-p150-wt, *** *p* = 0.000855, Figure 6B), in contrast with the mutant vectors it remained unaltered (Figure 6). Correlation analysis between the expression levels of ADAR1, *pri-miR-Let-7d*, and *Let-7d* after ADAR1 overexpression did not show significant differences (data not shown). Moreover, IPF fibroblasts were treated with Pentostatin, an inhibitor of ADAR, and downregulation of *pri-miRLet-7d* (*p* < 0.0001) expression was observed with an upregulation of *Let-7d* (*p* < 0.0001) (Appendix A).

### 2.5. Expression of COL3A1 and SMAD2 Targets of Let-7d in IPF

The expression of *COL3A1* and *SMAD2* mRNA targets of *Let-7d* potentially involved with the progression of IPF was analyzed in lung fibroblasts of control and IPF samples. The *Let-7d* target, *COL3A1* obtained by in silico analysis, was upregulated (** *p* = 0.006, Figure 7A) in IPF when *Let-7d* expression was decreased. Over-expression of ADAR1p110 wt isoform showed a negative effect on the expression levels of *COL3A1* (*p* < 0.05) and *SMAD2* (*p* < 0.05) (Figure 7B,C) in both control and IPF fibroblasts, however, ADARp150 wt showed an increased effect in the expression of *COL3A1* in normal and IPF fibroblasts (*p* < 0.0001) (Figure 7B). However, expression levels of *SMAD2* did not show changes by ADARp150 (Figure 7C).

## 3. Discussion

The ADAR1-p110 isoform is down-regulated both transcript and protein level, while the ADAR1-p150 isoform is overexpressed in IPF fibroblasts, even though at protein ADAR1-p150 level, there are no differences. This discrepancy may be due in part to the fact that the messenger RNA encoding ADAR1p150 has IRES-type elements in its sequence, which in pathological or stress conditions could respond to the presence of IRES transactivating factors, such as the polypyrimidine track binding protein (PTBP1). This results in the use of different translation starts, producing different isoforms of ADAR1 including ADARp110 [25]. Additionally, the distribution of these proteins is different in control fibroblasts; ADAR1-p110 is found mainly in the nucleolus, while in fibrotic cells, it is found in the nucleoplasm (Figure 2A–C). Previous reports showed that nucleolar ADAR2 location decreases its editing activity, and we hypothesize that the same regulation would be present in ADAR1 since both ADARs have dsRNA-binding motifs (DRBMs) and binding to rRNA [26]. Then, we propose that the ADAR1-p110 isoform is not correctly performing the editing in controls. In the case of fibrotic fibroblasts, this isoform is in the nucleoplasm, a region in which it has been demonstrated that the catalytic activity of this protein is executed [20], and even though the fibrotic fibroblasts have a diminished expression of the ADAR1-p110 isoform, their editing activity is increased. A report has mentioned that the ADAR editing activity is independent of their protein levels and expression levels [21], which supports our hypothesis that the ADAR1-editing activity depends mainly on its subcellular location. The expression levels of the ADAR1 isoforms and their location indicate that the regulatory mechanisms of each isoform are specific and complex. Differential locations for each ADAR1 isoform make us think that ADAR1-p110 is responsible for editing *pri-miR-Let-7d* and that cytoplasmic ADAR1-p150 for editing *Let-7d* transcript.

When the basal expression of an ADAR1 isoform is low, as in the case of ADAR1-p110, and is exogenously overexpressed, its edit activity levels do increase. Hence, we could suggest the existence of a negative feedback mechanism, which is essential to maintain specific expression levels in ADAR1 proteins to achieve an optimal function of the cells. However, we observed that the edition activity significantly increased in both control and fibrotic primary cultures when transfected with the wild-type isoforms, suggesting the existence of other regulatory mechanisms involved in the catalytic activity of ADAR in IPF fibroblasts. Analysis in silico showed that *pri-miR-Let-7d* has more than 70% adenosines edited by ADAR1; it is considered hyper-edition. Hyper-edited primary transcript is accumulated and rapidly degraded by Tudor-SN and is a less efficient substrate for Drosha and Dicer [26,27,28]. Thus, ADAR1 may inhibit the processing of the *pri-miR-Let-7d* through the loss of its double-stranded secondary structure. Consequently, Dicer and Drosha do not recognize it, redirecting it to another target or its degradation [29,30,31,32]. This degradation or deficiency in the processing of the *pri-miR-Let-7d* induced by the hyper-edition of the same transcript could explain the decrease in the *Let-7d* mature transcript in IPF since a down-regulation was observed along with ADAR1 expression.

The *COL3A1* and *SMAD2* mRNAs, putative targets of *Let-7d*, were quantified in IPF and controls. High levels of *COL3A1* and *SMAD2* were corroborated in IPF. A decrease in *Let-7d* expression induces an increase in *COL3A1* and *SMAD2* expression, which are involved in the development of pulmonary fibrosis.

When ADAR1 isoforms are over-expressed, the expression levels of *COL3A1* and *SMAD2* mRNA are modified, ADARp-110 induces downregulation of *COL3A1* and *SMAD2*, and interestingly ADARp-150 induced an increase in *COL3A1* on both normal and IPF fibroblasts. In this sense, we propose that the ADAR1-p150 isoform is editing *pri-miR-Let-7d*, avoiding its maturation to *Let-7d*. However, this has no effect on *SMAD2*.

To demonstrate the direct relationship between ADAR1 activity and the expression of *Let-7d*, we performed the experiment treating fibrotic fibroblasts with pentostatin, an inhibitor of ADAR activity (Appendix A). We were able to observe that *Let-7d* levels increased and *pri-miR-Let-7d* levels decreased, which demonstrates that ADAR1 activity directly influences the inhibition of maturation of *pri-miR-Let-7d*. These observations correlate with what is shown in Figure 4, where we observe a decrease in ADAR activity, which is reestablished with the transfection of the ADAR1 isoforms.

ADAR1-p110 and ADAR1-p150 regulate the processing of the *pri-miR-Let-7d* and mature *Let-7d* in such a way that the sub-expression of the latter deregulated, in turn, the expression of *COL3A1* and *SMAD2*, proteins involved in the development of idiopathic pulmonary fibrosis. In summary, the regulation of biosynthesis and the processing of *Let-7d* using the ADAR1 isoforms should be the subject of new studies and, in the future, could be considered important therapeutic targets for idiopathic pulmonary fibrosis.

Our study has several limitations. At first, we did know how many ADAR1 editing sites have *pri-miR-Let-7d* and if these sites could be promoting its maturation or its degradation, which has been reported for *pri-miR-142*, whose degradation is ADAR1-mediated-hyper-edition [31]. On the other hand, we did not explore the role of *Let-7d* in the conformation of the nucleolus in normal fibroblasts compared with fibrotic fibroblasts because it has been reported that *Let-7d* takes part in the nucleolar organization in mouse lung fibroblasts (MLg and MFML4), mouse lung epithelial cells (MLE-12), mouse mammary gland epithelial cells (NMuMG), and human lung epithelial carcinoma cells (A549); participate in the epigenetic regulation of several genes [32]. Future work will be necessary to investigate the role of ADAR-2 as a possible regulator of *Let-7d* and determine the role of ADAR-3, an isoform without catalytic activity but with a role of inhibitor of ADAR1 and 2 because it can form inactive dimers in combination with both proteins.

## 4. Materials and Methods

### 4.1. Cell Culture

The study population consisted of primary fibroblasts isolated from control (*n* = 10) or IPF (*n* = 10) lung tissue biopsies collected in the frame of the Mexican and the European IPF. All cells were cultured with Ham’s/F-12 medium (Thermo-Fischer, San Francisco, CA, USA) supplemented with 10% Fetal Bovine Serum (Thermo-Fischer, San Francisco, CA, USA). Patient characteristics are shown in Appendix A.

### 4.2. RNA Isolation, Reverse Transcription, Quantitative PCR

The total RNA of fibroblast and mice lung tissue was extracted with Trizol (Thermo-Fischer, San Francisco, CA, USA) and quantified using the Nanodrop system (Thermo-Scientific, San Francisco, CA, USA). The cDNA synthesis was performed using total RNA and the High-Capacity cDNA Reverse Transcriptase kit (Applied Biosystems, San Francisco, CA, USA). Quantitative real-time PCR reactions were performed using SYBR^®^ Green (Applied Biosystems, San Francisco, CA, USA) and TaqMan gene expression Real-Time PCR system (Applied Biosystems, San Francisco, CA, USA) on the Step-One-plus Real-time PCR system (Applied Biosystems, San Francisco, CA, USA) according to the manufacturer’s instructions as previously described [32]. The housekeeping gene HPRT was used to normalize gene expression. All primer pairs and TaqMan assay used are described in Appendix A.

### 4.3. Western Blot

Western blots of total protein were performed as previously described. Proteins were isolated using RIPA lysis buffer (Santa-Cruz Biotechnology, Dallas, TX, USA). Proteins were quantified using the Bradford method (Pierce, CA, USA) in a Beckman DU640 spectrophotometer [32].

All samples were incubated at 95 °C for 5 m with Laemmli buffer, followed by SDS-PAGE. Samples were transferred to nitrocellulose membranes, which were blocked for 1 h at room temperature with 5% fat-free milk in PBS-T 0.05%. Membranes were incubated overnight with a primary antibody at 4 °C (ADAR1, sc-271854; Lamin B, sc-17810, Santa-Cruz Biotechnology), washed with 0.05% PBS-T, and incubated for 1 h at room temperature with an HRP-linked secondary antibody, which was detected with the Super Signal West Dura Extended Duration Substrate detection solutions (Thermo-Scientific, San Francisco, CA, USA). The signal was detected and analyzed with the Chemi-Doc technology system (Bio-Rad, Hercules, CA, USA).

### 4.4. Immunofluorescence

Immunofluorescence was performed as described previously [12]. Briefly, control and IPF fibroblasts were seeded on glass slides, incubated for 24 h in RPMI at 37 °C, 5% CO_2_, then fixed with cold methanol, blocked for 20 m with 1× universal blocker (Biogenex, Fremont, CA, USA), and incubated with primary antibodies against ADAR1 (sc-271854) and fibrillarin (sc-374022) (Santa-Cruz Biotechnology, Dallas, TX, USA). Slides were incubated with secondary antibodies coupled to a fluorophore (Jackson-ImmunoResearch, Philadelphia, PA, USA) and analyzed using an FV-1000 confocal microscope (Olympus, Japan).

### 4.5. Generation of ADAR1-p110 Deletion Mutant

ADAR1-p150 variant was PCR cloned with Platinum-Pfx Polymerase (ThermoFisher, San Francisco, CA, USA) following standard protocols and using ADAR1-p150 wt [18,22] as a template. PCR-Products were gel purified and digested with BamHI (#R0136S NEB, Ipswich, MA, USA) and XbaI (#R0145S, NEB) and ligated overnight at 16 °C into linearized pcDNA3.1-Flag-HA (RRID:Addgene_52335, Cambridge, MA, USA). ADAR1-p110 mutant (catalytically inactive) was generated using ADAR1E912A [22,30] as a template and PCR cloned. All constructs were sequence verified.

### 4.6. ADAR Catalytic Activity Assay

Editing of RNA in control fibroblasts compared with IPF fibroblasts was measured using the RESS-qPCR [30] as follows. As an edition template, *APOBEC3D*, the ADAR1-mediated editing target, was selected. cDNA of 6 control lines and 6 fibrotic lines (2 μL cDNA per sample) was amplified by qPCR (Appendix A), by serial dilutions, and the efficiency of the primers for *APOBEC3D* wt and *APOBEC3D* edit was tested, and in a final volume of 10 μL (5 μL #1176202K SYBR GreenER Super Mix Life Technologies, San Francisco, CA, USA. 0.2 μL of primers, calibrated to 10 μL with nuclease-free water). Edition ratios were calculated using the values of the efficiency of the primers in the following way:(1)$$\mathrm{Relative}\;\mathrm{ratio}\;\mathrm{of}\;\mathrm{RNA}\;\mathrm{edition}=\frac{E^{-ct\;Edit}}{E{\;}^{-ct\;\mathrm{wt}}}$$

Control and IPF primary cultures were transfected with overexpression constructs for ADAR1-p110 wt, ADAR1-p110 mutant, ADAR1-p150 wt, and ADAR1-p150 mutant, as well as the empty vector. Cells were treated with 0.5 μM of pentostatin (#sc-204177 Santa-Cruz Biotechnology, Santa Cruz, CA, USA) 24 h post-transfection to inhibit the catalytic activity of Adenosine Deaminase (ADA) in order to increase the specificity of this assay. Cells were collected 48 h post-transfection and used for the catalytic activity test.

The formation of inosine (from the hydrolysis of adenosine) was measured in aliquots of the proteins extracted from the cells transfected with specific plasmids that were used for each isoform wild-type and mutant of ADAR1, the colorimetric assay Adenosine Deaminase Activity Assay Kit (#K321-100 Biosvision, Milpitas, CA, USA) was used. The protein extract of the cells was prepared by adding 150μL of cold ADA 1× assay buffer with protease inhibitor (#14865751 Thermo-Scientific, San Francisco, CA, USA). Dilutions of inosine standard were used as a standard curve. As a negative control, 50 μL of 1× ADA Assay Buffer was used. The reading plate was incubated at 37 °C for 5 m, after which it was read at a wavelength of 293 nm using the Infinite 200 plate reader (Tecan, Männedorf, Switzerland) at time points 0, 10, 20, and 40 m. All data were analyzed using the following formula:(2)$$\mathrm{Editing}\;\mathrm{activity}\;\mathrm{of}\;\mathrm{ADAR}\;\mathrm{per}\;\mathrm{sample}=\frac{B}{\Delta T\;\cdot \;\mu g\;de\;prote\text{í}na}\cdot \;DF=\mathrm{nmol}/\min/\mu g\;=\mathrm{mU}/\mu g$$

Inosine standard curves were set at 4 time points for the analysis.

### 4.7. Selection of Target mRNA

In silico analysis was performed to investigate *Let-7d* targets, using miRWalk, miRanda, and TargetScan algorithms. Obtained readouts were compared to select the top 250 mRNA with differential expression in IPF, and *Collagen III alpha 1* (*COL3A1*) and *SMAD2* were selected.

### 4.8. Pentostatin Treatment in IPF Fibroblasts

IPF fibroblasts were treated with 0.5 μM of pentostatin (#sc-204177 Santa-Cruz Bio technology, Dallas, TX, USA). At 24 h after, *Let-7d* and *pri-mir-Let-7d* expressions were determined by real-time PCR as described before.

### 4.9. Statistical Methods

All data follow a normal distribution, the Student´s test was used to compare RNA expression, and values are expressed as mean ± SD. All statistical analyses were performed using GraphPad Prism (San Diego, CA, USA).

## Figures and Tables

**Figure 1 ijms-23-09028-f001:**
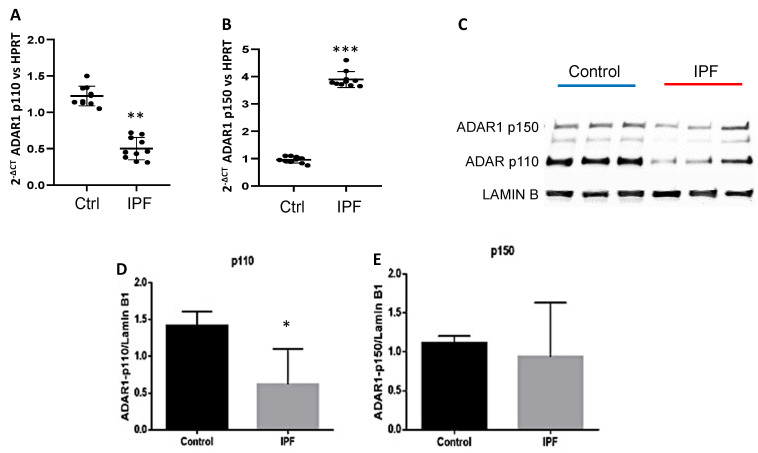
Expression levels of ADAR1-p110 and ADAR1-p150 in control compared with fibrotic fibroblasts. (**A**) ADAR1-p110 is significantly down-regulated in IPF fibroblasts (** *p* < 0.0001). (**B**) ADAR1-p150 is significantly overexpressed in IPF fibroblasts (*** *p* < 0.0001). (**C**) Analysis of the protein levels of the isoforms of ADAR1 in total protein extracts corroborated the decrease in ADAR1-p110 (* *p* < 0.05), whereas ADAR1-p150 did not show significant differences. (**D**) Densitometric analysis of the protein levels of ADAR1 (**C**) corroborates the decrease in ADAR1-p110 in fibrotic cells and no difference in ADAR1-p150 (E).

**Figure 2 ijms-23-09028-f002:**
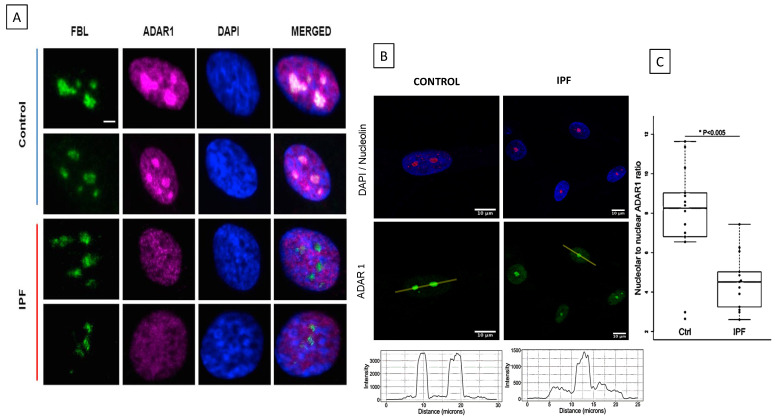
Immunolocalization of ADAR1 in the nucleus on control and fibrotic pulmonary fibroblasts. (**A**) Fibrillarin (FBL, green; lane 1) was used as a specific marker for the nucleolus. ADAR1 was observed in magenta color (lane 2). Nuclei-stained with DAPI (lane 3). In control fibroblasts, ADAR1 is in the nucleolus and nucleoplasm (lane 2), whereas in fibrotic fibroblasts, the distribution of ADAR1 occurs mainly in the nucleoplasm. Merge image for FBL, ADAR1 and DAPI (lane 4). Bar = 5 μm. (**B**) ADAR1 (shown in green) seems to be predominantly detected in the nucleolus in both control and IPF fibroblasts, as shown by its similar distribution in the nucleolus marker Nucleolin (shown in red). However, ADAR1 expression is reduced in the nucleoli of IPF fibroblasts compared to control fibroblasts; in contrast, its expression is enriched in the nucleoplasm, as shown in the intensity profile (yellow line, intensity values plotted below the micrographs). (**C**) The ratio of ADAR1 nucleolar to nuclear expression is significantly higher in control fibroblasts than in IPF fibroblasts (mean of ratios in Ctrl group = 7.84 {*n* = 14}, the mean of ratios in the IPF group = 4.52 {*n* = 13}, *p* = 0.0015, Wilcoxon rank-sum test).

**Figure 3 ijms-23-09028-f003:**
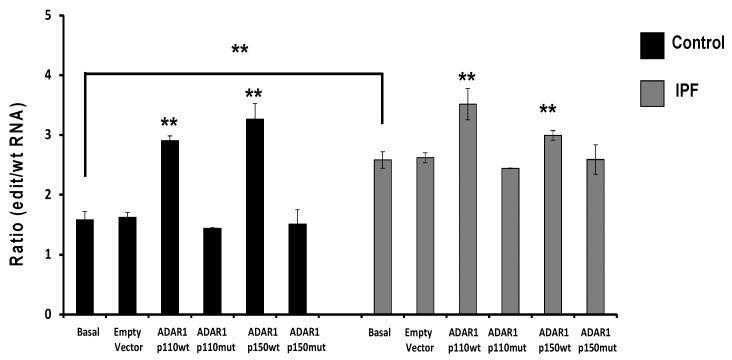
The catalytic activity of ADAR1 on *APOBEC3D* in control fibroblasts compared with fibrotic fibroblasts. ADAR1 catalytic activity in basal conditions in IPF fibroblasts concerning control fibroblasts (** *p* = 0.0018). Editing activity increase is shown as the ratio of the edition divided by the ratio of wild-type (*n* = 6). The catalytic activity is increased after transfection with the catalytically active plasmids, whereas the mutant plasmids do not modify the activity. In control fibroblasts (black bars), ADAR1 p110 wt shows a significant increase in editing activity (** *p* = 0.003737). Remarkable increase in editing activity is observed with the ADAR1-p150 isoform (** *p* = 0.000139), while the greatest increase the editing activity in IPF fibroblasts are observed after over-expression of the ADAR1-p110 (** *p* = 0.007030). An increase in ADAR1-p150 wt activity was also significant (** *p* = 0.009158).

**Figure 4 ijms-23-09028-f004:**
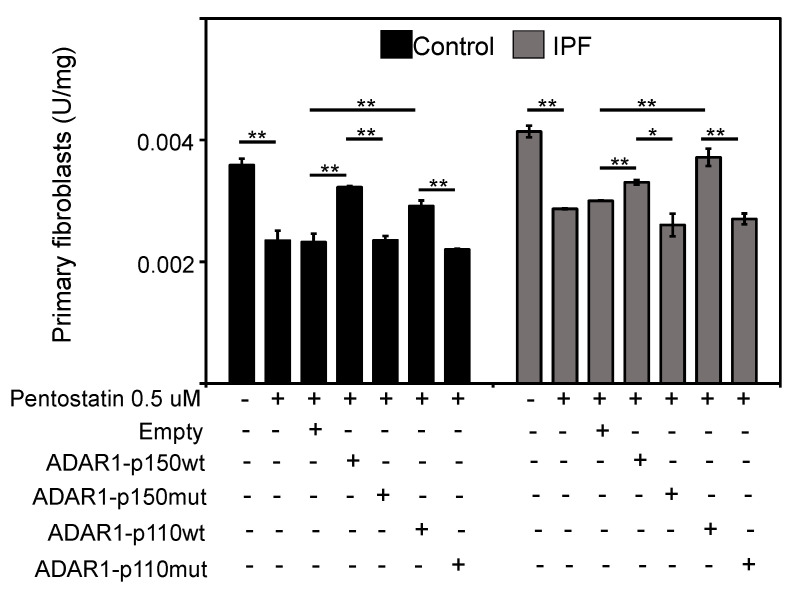
The specific catalytic activity of ADAR1 is increased in primary fibroblasts of IPF. Pentostatin, as a specific ADA inhibitor, significantly decreased the catalytic activity in control fibroblasts (** *p* = 0.0059) and IPF fibroblasts (** *p* = 0.0014). Over-expression of ADAR1-p150 wt (** *p* = 0.0058) and ADAR1-p110 wt (** *p* = 0.0018) significantly increased the catalytic activity of ADAR in control fibroblasts. Over-expression of ADAR1-p150 (* *p* = 0.8) and ADAR1-p110 mutant (* *p* = 0.72) did not induce significant changes in the catalytic activity in the control fibroblasts. ADAR1-p150 wt over-expression in IPF fibroblasts significantly increases its activity (** *p* = 0.0042) like the over-expression of ADAR1-p110 (** *p* = 0.0067). Over-expression of ADAR1 mutant plasmids does not modify the activity. ADAR1-p150 mutant (* *p* = 0.2) and ADAR1-p110 mutant (* *p* = 0.02).

**Figure 5 ijms-23-09028-f005:**
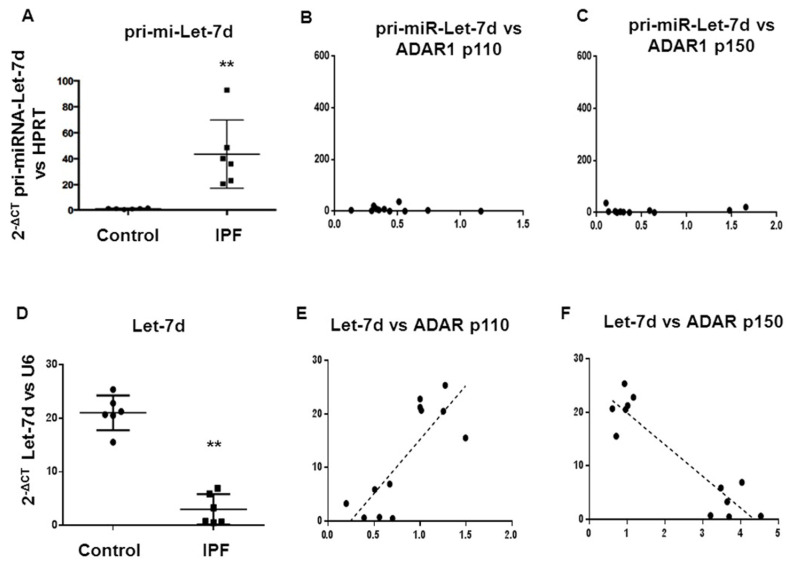
*pri-miRNA-Let-7d* is over-expressed in IPF fibroblasts, and *Let-7d* is downregulated in IPF fibroblasts and fibrotic mice lugs. The expression of pri-miRNA-*Let-7d* is higher in IPF fibroblasts than in control fibroblasts (*p* = 0.05) (**A**). The correlation of ADAR1 isoforms does not show significant differences between ADAR1-p110 (**B**) and ADAR1-p150 (**C**). The expression of *Let-7d* is lower in IPF fibroblasts compared to the expression levels of control fibroblasts (** *p* = 0.0028) (**D**). *Let-7d* displayed a negative correlation with ADAR1-p150 (*p* < 0.0001) (**E**) and a positive correlation with ADAR1-p110 (*p* = 0.0014) (**F**).

**Figure 6 ijms-23-09028-f006:**
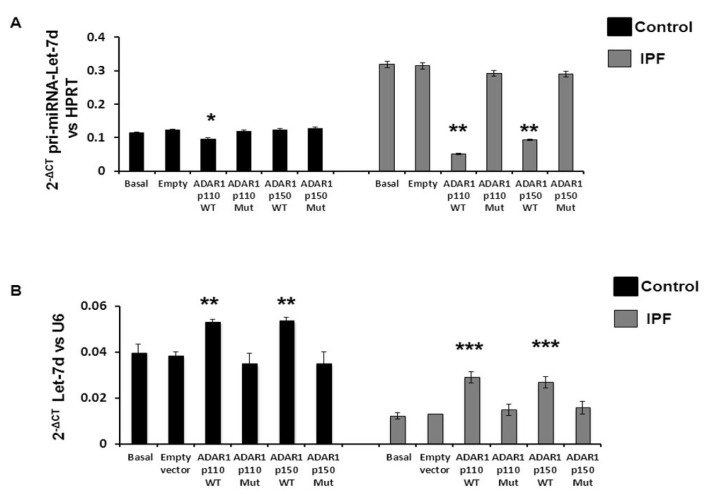
The expression of *pri-miR-Let-7d* is modified by the over-expression of ADAR1-p110 wt and ADAR1-p150 wt. In (**A**), the over-expression of ADAR1-p110 wt in control fibroblasts significantly decreases the expression of *pri-miR-Let-7d* (* *p* = 0.054479, black bars). In IPF fibroblasts, both isoforms promote the decrease in *pri-miR-Let-7d*. ADAR1-p110 wt (** *p* = 0.0002108, gray bars) and ADAR1-p150 wt (** *p* = 0.00038554) in a higher proportion compared to the control fibroblasts. The expression of *Let-7d* in (**B**) was increased in the fibroblasts transfected with ADAR1. In control fibroblasts transfected with ADAR1, the expression of *Let-7d* is increased significantly ADAR1-p110 wt (** *p* = 0.001477, black bars) and ADAR1-p150 wt (** *p* = 0.001946, black bars). In IPF fibroblasts, the over-expression of catalytically active isoforms promotes a significant increase in *Let-7d*. ADAR1-p110 wt (*** *p* = 0.000151, gray bars) and ADAR1-p150 wt (*** *p* = 0.000855, gray bars).

**Figure 7 ijms-23-09028-f007:**
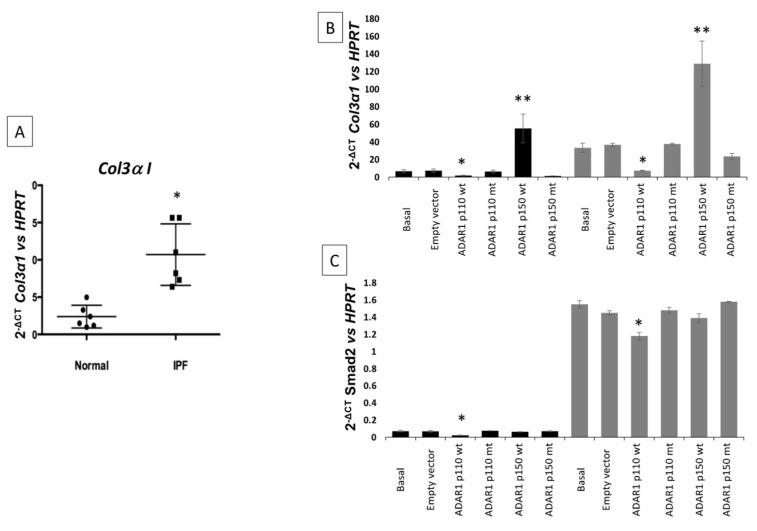
Over-expression of *COL3A1* in IPF fibroblasts (black square) vs Normal fibroblasts (black circle) (* *p* < 0.05) (**A**), Over-expression of ADAR1p110 wt isoform showed a negative effect on the expression levels of *COL3A1* (*** < 0.05) and *SMAD2* (* *p* < 0.05) (**B**,**C**) in both control and IPF fibroblasts; however, ADARp150 wt showed an increased effect in the expression of *COL3A1* in normal and IPF fibroblasts (** *p* < 0.0001) (**B**). The expression of *SMAD2* did not show changes by ADARp150 (**C**).

## Data Availability

Not applicable.

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
