# Peer review of "ADAR1 Isoforms Regulate Let-7d Processing in Idiopathic Pulmonary Fibrosis"

_ijms, 2022, doi:10.3390/ijms23169028_

Round 1

Reviewer 1 Report

The manuscript by Dina-Piaz et al explores the role of ADAR1 isoforms and Let-7d which could have synergistic effects in the development of idiopathic pulmonary fibrosis and manipulating the expression patterns of these molecules could be used as a therapeutic intervention for IPF.

The manuscript is well written. There are a couple of typos which needs correction and I have minor concerns.

Fig 2A: The fluorescence intensity of ADAR1 was shown. Can authors comment whether it was p110 or p150 isoform?

Line 144, should be fig 5e

Fig 1: PCR data shows p110 downregulated while p150 shows upregulated. while western blot data protein level, p150 shows non significant donwregulation or no change. Can authors explain this discrepancy?

Fig 3: Label A and B

Fig 5: positive and negative correlation with p110 and p150 should be discussed.

Fig 5E: What was the purpose of showing Bleomycin mice model? The authors must show that the model worked and should discuss the results.

Table 1: Should be German subjects.

Line 291, should be experimental pulmonary fibrosis.

Line 298-300, please revise because its confuses the reader.

The conclusion is vague and does not accurately concludes the results. Whether ADAR regulates Let7d directly or indirectly, studies must be confirmed using inhibitors.

Author Response

Responses to reviewer 1.

Dear Reviewer

We greatly appreciate your interest and suggestions to improve our work, I hope that our answers leave you satisfied.

The manuscript by Dina-Piaz et al explores the role of ADAR1 isoforms and Let-7d which could have synergistic effects in the development of idiopathic pulmonary fibrosis and manipulating the expression patterns of these molecules could be used as a therapeutic intervention for IPF.

The manuscript is well written. There are a couple of typos which needs correction and I have minor concerns.

Fig 2A: The fluorescence intensity of ADAR1 was shown. Can authors comment whether it was p110 or p150 isoform?

The ADAR1 antibody (sc-271854, Santa-Cruz Biotechnology) recognizes both isoforms. ADAR1 has two isoforms generated by the alternative promoter, ADAR1- p110 and ADAR1-p150. ADAR1-p110 is the amino-terminally truncated isoform located in nuclear regions and constitutively expressed. ADAR1-p150 is the full-length isoform, which is interferon-inducible and is preferentially located in cytoplasmic regions [14,21].

Line 144, should be fig 5e Figure was eliminated

Fig 1: PCR data shows p110 downregulated while p150 shows upregulated. while western blot data protein level, p150 shows non significant donwregulation or no change. Can authors explain this discrepancy?

This discrepancy may be due in part to the fact that the messenger RNA encoding ADAR1p150 has IRES-type elements in its sequence, which in pathological or stress conditions could respond to the presence of IRES transactivating factors, such as the polypyrimidine track binding protein (PTBP1) This results in the use of different translation starts, producing different isoforms including ADARp110. (25). Reference 25 was added.

Fig 3: Label A and B.

It is a single figure, and we made a mistake in the description, and we have already corrected the error. Thanks for your observation.

Fig 5: positive and negative correlation with p110 and p150 should be discussed.

Your suggestion was corrected, thank you.

Fig 5E: What was the purpose of showing Bleomycin mice model? The authors must show that the model worked and should discuss the results.

We greatly appreciate this observation and decided to withdraw the bleomycin experiment from the work as we consider that it is not relevant to this article.

Table 1: Should be German subjects.

The text in the table has been corrected, thanks.

Line 291, should be experimental pulmonary fibrosis.

We removed that paragraph.

Line 298-300, please revise because its confuses the reader.

Your suggestion was corrected, thank you.

The conclusion is vague and does not accurately concludes the results. Whether ADAR regulates Let7d directly or indirectly, studies must be confirmed using inhibitors.

We performed the experiment treating fibrotic fibroblasts with pentostatin, an inhibitor of ADAR activity, and we were able to observe that Let7d levels increased and primlet7d levels decreased, which demonstrates that ADAR1 activity directly influences the inhibition of maturation of primLET7d, which correlates with what is observed in figure 4, where we observe a decrease in ADAR activity, which is reestablished with the transfection of the ADAR1 isoforms.

We redraft our discussion.

Sincerely

Dr. Victor Ruiz

Reviewer 2 Report

In this manuscript, the authors investigated ADAR1 expression in IPF as well as the effect its overexpression or downregulation on the development of IPF. Authors have done a good work but unfortunately the quality of their work was lost due to poor presentation. Here are some possible suggestions for the authors to improve their manuscript.

Major:

1- Title: the title shouldn’t be all uppercase letters, also it should be modified to reflect the performed work.
2- Authors affiliation’s should be in  English.

3- The number of control and IPF lung samples need to be increased and should be from the same race if possible; or at least 5 samples from each race and it will be interesting if the authors indicate if there is a difference in the expression levels of ADAR1 between Mexicans and German patients.
4- All figures should be consistent, also figure 2 images resolution need to be improved and the expression in figure 2D is unclear.

Minor:
1- Tables 1 and 2 should be moved to the supplementary materials and revised for its accuracy.

Author Response

Responses to reviewer 2.

Dear Reviewer

We greatly appreciate your interest and suggestions to improve our work, I hope that our answers leave you satisfied.

Comments and Suggestions for Authors

In this manuscript, the authors investigated ADAR1 expression in IPF as well as the effect its overexpression or downregulation on the development of IPF. Authors have done a good work but unfortunately the quality of their work was lost due to poor presentation. Here are some possible suggestions for the authors to improve their manuscript.

Major:

1- Title: the title shouldn’t be all uppercase letters, also it should be modified to reflect the performed work.

We listened to your suggestion, thank you very much

2- Authors affiliation’s should be in  English.

We listened to your suggestion, thank you very much

3- The number of control and IPF lung samples need to be increased and should be from the same race if possible; or at least 5 samples from each race and it will be interesting if the authors indicate if there is a difference in the expression levels of ADAR1 between Mexicans and German patients.

We increased the number of samples from both Mexican and German patients to 5 per group. We verified that there were no differences in the basal levels of ADAR1 expression between both control and fibrotic fibroblasts. Comparisons between German and Mexican samples were added as supplementary figures (figure 1S).

4- All figures should be consistent, also figure 2 images resolution need to be improved and the expression in figure 2D is unclear.

We improved the resolution of the images and decided to eliminate immunohistochemistry in human histological sections.

Minor:
1- Tables 1 and 2 should be moved to the supplementary materials and revised for its accuracy.

Corrected. The tables were sent as supplementary figures Table S1 and Table S2.

Sincerely

Dr. Victor Ruiz

Round 2

Reviewer 1 Report

The authors have satisfactorily answered all the response to review comments.  I have no further comments.

Reviewer 2 Report

The authors improved the quality of the manuscript and responded to my questions.